# Dairy Sheep Grazing Management and Pasture Botanical Composition Affect Milk Macro and Micro Components: A Methodological Approach to Assess the Main Managerial Factors at Farm Level

**DOI:** 10.3390/ani12192675

**Published:** 2022-10-05

**Authors:** Andrea Cabiddu, Sebastian Carrillo, Salvatore Contini, Simona Spada, Marco Acciaro, Valeria Giovanetti, Mauro Decandia, Luigi Lucini, Terenzio Bertuzzi, Antonio Gallo, Lorenzo Salis

**Affiliations:** 1Agris Agricultural Research Agency of Sardinia, Loc. Bonassai, Olmedo, 07040 Sassari, Italy; 2Facultad de Estudios Superiores Cuautitlán, National Autonomous University of Mexico, Mexico City 54714, Mexico; 3Department for Sustainable Food Process, Catholic University of the Sacred Heart, Via Emilia Parmense, 84, 29122 Piacenza, Italy; 4Department of Animal Science, Food and Nutrition (Diana), Catholic University of the Sacred Heart, Via Emilia Parmense, 84, 29122 Piacenza, Italy

**Keywords:** pasture-based diet, fatty acid profile, vitamin, phenolic compounds, colour, dairy sheep

## Abstract

**Simple Summary:**

Studies on the management factors that affect milk components at the farm level are important for understanding how to transfer the results from experimental study. Plant phenological stages and partially fresh herbage intakes affect the lactose and milk fatty acid profile. The botanical composition of the grassland partially affects the milk’s phenol content. A few small relationships between plant phenols and milk colour could be of interest to explain the changes in milk colour parameters.

**Abstract:**

The fatty acid profile, vitamins A and E, cholesterol, antioxidant power colour and the phenols profile of Sarda sheep milk from 11 commercial sheep flocks managed under permanent grassland were investigated. In each farm, the structural and managerial data and milk samples were collected during four periods (sampling dates, SD): January, March, May, and July. Data from the milk composition (fat, protein, casein, lactose, and somatic cell count), 68 fatty acids, 7 phenols, 1 total gallocatechin equivalent, ferric reducing antioxidant power, vitamins A and E, cholesterol, degree of antioxidant protection, and the colour (b *, a * and L *) were analyzed by multivariate factorial analysis using a principal component analysis approach. A proc mixed model for repeated measurement to point out the studied factors affecting significant macro and micro milk composition was also used. Only the first five components were detailed in this paper, with approximately 70% of the explained variance detected. PC1 presented the highest positive loadings for milk lactose, de novo FA synthesis and the BH intermediate, whereas OBCFA had negative loadings values. The PC2, LCFA, UFA, MUFA, vitamins E, and DAP showed positive loadings values, while SFA had a negative value. The PC3 showed a high positive loading for total phenols and non-flavonoids. PC4 presented a high positive loading for the milk macro-composition and negative values for n-3 FAs. The PC5 is characterized by high positive loadings for the a * and L * colour parameters whereas negative loadings were detected for the milk flavonoids content. These preliminary results could help to establish future threshold values for the biomarkers in milk sourced from grazing dairy sheep in natural, permanent pasture-based diets.

## 1. Introduction

Consumers are showing growing interest in the methods of production of their food as well as in environmental issues. Dairy and meat products sourced from pasture-based diets are associated with positive characteristics since it is well known that grazing confers specific organoleptic features on the dairy products with high healthiness and sensorial value [1]. Of course, some ambiguity around the terminology of “grass-fed” vs. “pasture-based” has occurred: whereby “grass-fed” is an indication of the proportion of grass in fresh or ensiled feed, whereas “pasture-based” implies that the sheep are outdoors grazing [2]. In addition, while food safety concerns increase with the food supply chain length, the sharing of knowledge and interest among stakeholders decreases, with deleterious effects on trust and the understanding between farmers, processors, retailers, and consumers. In sheep farms, an increase in grazing and self-produced hay and feedstuff could reduce the feeding cost for purchased feed. Pasture-based feeding is sustainable, safe and delivers high quality products and a range of ecosystem services [3], related to milk production in local small ruminant breeds with high levels of grazing. In this context, farmers consider permanent pastures to have a strong link with the territory origin (heritage and identity) as also reported by [4]. Finally, consumers are prepared to pay more money for milk from pasture-based production systems, but dairy companies often say that it is difficult to obtain the expected extra value from the market [5].

In this context dairy products sourced from a pasture-based diet, could represent extra added value. In Sardinia, natural pasturelands represent the main source of animal feed, covering more than one million hectares, about 87% of the total forage area of the island [6,7]. The dairy sheep farming system is mainly characterized by semi-extensive farms with a high use of natural pasture and forage crops, and contributes 66% of the total national milk sheep production [6]. Sardinian sheep farms provide cheese with a protected denomination origin (e.g., pecorino Romano, pecorino Sardo, and Fiore Sardo) as also reported by [1] in a similar context.

However, in the last decades, in the face of low sheep milk prices, more and more intensive systems are being adopted in sheep farms with a high selection rate; using large amounts of concentrate and dried fodder, with less and less grazing activity, resulting in natural pasture abandonment, particularly in less favored areas [8,9].

Previous papers have demonstrated that in Sardinia, sheep diets based on forage crops and natural pasture with moderate supplementation can increase milk polyunsaturated fatty acid (PUFA) and conjugated linoleic acid (CLA) content and, consequently, the dairy products’ healthiness [10,11,12,13]. This is strongly impacted by the plants’ phenological stage and season, since herbage fatty acids (FA), the precursors of milk PUFAs, are high in the leaves of plants during the growing stage in the wet season [14]. Different feeding strategies that improve sheep milk quality have been tested under controlled conditions and reviewed recently; these considered zero grazing vs. full grazing crops vs. different fat supplementation [10,12,13,15].

In the Mediterranean basin, sheep milk production is seasonal (with herbage availability in late autumn-winter and early spring and a shortage in summer), with a short lambing period (November–December). In this context, a few constraints occur both for farmers (low pasture forage production in winter) and cheese factories (milk production mainly concentrated in winter and early spring). On the other hand, in late spring, grazing animals appear more economically sustainable (total dry matter intake from fresh herbage covers about 85% of the animals’ requirements) with a lowering of feeding costs. This implies that several concerns exist when implementing the authentication process for pasture-based feed in sheep dairy products [16]. Recent papers underlined, at the experimental level, that several tools could be implemented to increase the authentication methods of dairy products, such as milk alkanes and fatty acid profile detection [17,18]. In the dairy sheep farming system, data on the influence of permanent pasture on milk quality traits are scanty, in particular regarding carotenoids, colour and phenol contents as reported in a recent review [13], at both controlled conditions and at farm level. To our knowledge, the effect of supplementation on milk components in sheep grazing permanent pasture are reported only partially, because the authors reported only the effect on the macro components and fatty acid profile [6], or the macro components, vitamins, and volatile organic compounds [1] and Valdivielso et al. cited by [1].

Nowadays, the availability of a large number of milk chemical components that can be detected, and the experimental factors analyzed, give some concern about the best statistical approach to implement ”on farm” the best managerial practice to improve the milk quality traits. The evaluation of the simultaneous variation in milk composition due to different variables could help us to increase knowledge of the factors affecting milk’s nutritional and sensorial properties; the use of multivariate statistical analysis allows one to capture the covariance structure of complex patterns of variables and create a simpler interpretation of the original multivariate system [19]. Recently, several papers were published on the effect of livestock systems on sheep milk fatty acid profile collected at the farm level, [20,21] considering only partially the farm managerial factors and often without any references to the grassland’s botanical composition detected during the survey.

The aim of this paper is to investigate, in extensive dairy sheep farms, the main relationships between a few structural (stocking rate) and managerial (animal diet composition, pasture botanical composition, plant phenological stage, total supplementation) factors, and the milk’s macro-composition (fat, protein, casein, somatic cell count, lactose) and micro-composition (FA profile, vitamins, colour, phenols and other antioxidant parameters). The study aims to assess a data methodological approach in a case study at the commercial farm level, to aggregate a large number of variables detected in sheep milk composition into fewer latent structures, namely the principal components (PC). These latent structures are subsequently related to some managerial factors to point out the milk biomarkers that could be useful to test the link between dairy products and the territory of origin. This approach is a first step in providing a small dataset at the farm level, and later needs to be validated with a large database that takes into account the effect of years. Our basal hypotheses to test at farm level are: (i) does the grassland’s botanical composition and plant phenological stage interact with feeding supplementation to affect the milk’s macro and micro chemical composition? (ii) Is it possible to synthetize into a small group of factors a large number of variables (about 100) related to the milk’s macro- and micro-composition? 

## 2. Materials and Methods

### 2.1. Dairy Farming System and Pasture Botanical Characteristics

The study was performed in 2019, in 11 dairy sheep (Sarda breed) farms located in a hilly central area of Sardinia (historic region of Marghine) (300–700 m a.s.l.). The Sarda breed is a native medium-sized sheep (45 kg of live body weight), mainly raised on a semi-extensive system based on permanent crop grassland. The average milk production is 220 L in 180 days of lactation. In each farm, milk samples were collected during 4 sampling days (SD), in January, March, May and July 2019 when all of the sheep had 58, 98, 138 and 178 days in milk (DIM), respectively. The SD were carried out every 40 days to better link the milk yield and composition to the evolution of the botanical composition phenological stage and the fresh herbage availability during the lactation period of the dairy sheep in the Mediterranean basin. In total, 42 milk samples were collected, as on two occasions the samples were lost. At every SD, structural and managerial data were also recorded: Utilised Agricultural Area (UAA), flock size (number of sheep), and feeding management, such as hay and concentrate supplementation, access time to grazing per day, and the forage conservation systems. Animal diet composition was indicated as the intake of (i) grazed herbage in natural pasture (HeI); (ii) hay (HI); (iii) concentrate (CI); and (iv) standing hay. Pasture herbage intake (Hel) was estimated by the difference between the potential intake capacity of the animals at the different SD and the encumbrance provided by the other feedstuffs fed to the sheep [22]. The potential intake capacity was estimated according to the herd’s characteristics (milk yield and composition, and animal body weight) as detailed in the small ruminant nutrition system (SRNS) [23]. A complementary study was carried out to evaluate the botanical composition of the pasture grazed by the sheep on the same days as milk sample collection. In each SD, the herbage mass was determined by cutting the sward at 3 cm above ground level on a sampling area of 0.5 m^2^. The pasture botanical composition (BC) was detected by partitioning the herbage samples into Legumes, Grasses, Forbs species. The plant phenological growth stages (PPS) were detected at each SD: (i) Growing stage (GW, from 0 to 5 (Biologische Bundesanstalt, Bundessortenamt und CHemische Industrie) BBCH scale; Refs. [24,25] characterized by plants in active growth and with vegetative activity; (ii) flowering stage (FW, from 5 to 6 BBCH; Refs. [24,25] characterized by plants with inflorescence or flower buds visible until the end of flowering; (iii) maturity or senescence stage (MS, from 8 to 9 BBCH; Ref. [25] characterized by no vegetative activity, leaf fall and with most of the plants dead or dormant. The fresh forage was dried in an oven-dryer at 65 °C for three days to evaluate the proportion (%) of each partitioned group in terms of dry matter (DM).

### 2.2. Milk, Herbage and Feedstuff Chemical Characteristics

The ewes were milked twice a day (in the evening and morning) using automatic milking machines. In each SD, 1 bulk milk sample (1.5 L) deriving from the daily milking was taken from the tank of each farmhouse. Five sub samples were obtained from each farm: one for macro-composition, one for vitamins, cholesterol and the degree of antioxidant protection (DAP), one for the phenols profile, one for ferric reducing antioxidant power (FRAP), one for gallic acid equivalent detection (GAE), and one for the fatty acid (FA) profile. The fat, protein, casein, somatic cell counts (SCC) and lactose contents were determined by Milkoscan FT+, Foss Electric, (Hillerød, Denmark). Milk color was measured by Chroma Meter CR400/410 (Konica Minolta Sensing Japan) with a spectral sensitivity characteristic of the sensor and the spectral distribution of the xenon lamp. Three readings per sample were performed. The white calibration was performed with a white calibration plate where Y = 93.5, x = 0.3114, and y = 0.3190. The following parameters were detected: L * (light-ness; on a scale from 0 to 100, where 0 = black and 100 = white), a * (where −a * has a green color and +a * has a red color), and b * (where −b * has a blue color and +b * has a yellow color).

For the milk fatty acid profile detection, milk fat was obtained in agreement with [26]. Fatty acid methyl esters from milk fat (0.05 g of milk fat sample) were obtained according to the international standard (FIL-IDF, 1999), using base-catalyzed trans-methylation. Separation of FAMEs was performed [26]. To achieve a compromise between an acceptable separation of the cis and trans isomers (C16:1, C18:1 C18:2 and C18:3) and the time of analysis, one of the oven temperature programs proposed by [27] was used. Individual FAMEs were identified by comparison with a standard mixture of 38 FA methyl esters (Matreya Inc., Pleasant Gap, PA, USA). C18:1 isomers standards (C18:1t9, C18:2 t9t12, C18:1t11, Matreya Inc., Pleasant Gap, PA, USA) and CLA standards (CLAc9t11; CLAt10c12; CLAc9c11; CLAt9t11, Matreya Inc., Pleasant Gap, PA, USA) and published isomeric profiles [28] were used to identify the C18:1 isomers, C18:2 non conjugate isomers and the CLA isomers. Calibration curves with internal standards were performed to quantify fatty acid methyl esters (FAMEs) that are calculated as g FAME/g of milk fat and then expressed as g FAME/100 g of total identified FAMEs. The internal standard Me-C5:0 was used to quantify from Me-C4:0 to Me-C7:0, the internal standard Me-C9:0 was used to quantify from Me-C8:0 to Me-C10:0, the internal standard Me-C13:0 was used to quantify medium chain FAMEs from Me-C11:0 to Me-C17:0 and the internal standard Me-C19:0 to quantify long chain FAMEs from Me-C18:0 to Me-C26:0. The concentration of each internal standard, added to the milk fat sample prior to transmethylation, was 0.005 g in 0.050 g of milk fat.

Vitamin A, E and cholesterol were analyzed according to [29]. Briefly, aliquots (2 mL) of milk were digested with 2 mL of KOH (60% aqueous solution, *w*/*v*), 2 mL of 95% ethanol, 1 mL of NaCl (1% aqueous solution, *w*/*v*), and 5 mL of an ethanolic solution of pyrogallol (6%, *w*/*v*) added as an antioxidant. After digestion in a water bath at 70 °C, the suspension was cooled for 30 min, and 5 mL of an NaCl solution (1%, *w*/*v*) was added to prevent emulsification. The suspension was then extracted with 10 mL of n-hexane/ethyl acetate (9:1, *v*/*v*). The lower aqueous layer was extracted 3 more times, with 5 mL of n-hexane/ethyl acetate (9:1, *v*/*v*) each time. The pooled organic layers were evaporated with a rotary evaporator at 30 °C, and the dry sample was dissolved in 3 mL of methanol for HPLC. A sample volume of 20 µL was injected in HPLC equipment, previously filtered using a 0.20 µm PTFE filter. All determinations were carried out in duplicate. The degree of antioxidant protection (DAP), which represent a synthetic index, was calculated as the ratio between the amount of antioxidant element (e.g., α-tocopherol) and the element to be protected against oxidation (cholesterol) according to [30].

The original FRAP assay [31] protocol was modified to make the assay suitable for raw milk according to [32]. The FRAP reagent was prepared by mixing 300 mM sodium acetate buffer (pH 3.6) and 10 mM TPTZ (2,4,6-Tris(2-pyridyl)-s-triazine) in a 40 mM HCl and 20 mM iron (III) chloride solution in a volume ratio of 10:1:1, and the mixture was warmed to 37 °C in a water bath before use. Afterward, 900 µL of the prepared FRAP reagent was mixed with 30 µL of a diluted sample, and an absorbance at 593 nm was recorded after 5 min of incubation at 37 °C. A standard curve was constructed using iron sulfate heptahydrate (FeSO_4_·7H_2_O), and the data were expressed as millimoles of FeSO_4_ equivalents per kilogram of cheese. All determinations were carried out in duplicate. Gallic acid equivalents (GAE) were analyzed according to [33] using the Folin–Ciocalteu colorimetric method. Briefly, 100 µL of the diluted extract was mixed with 500 µL of a 0.2 N Folin–Ciocalteu reagent. Afterward, 400 µL of a sodium carbonate solution (7.5% aqueous solution, *w*/*v*) was added to the reaction mixture. The absorbance readings were taken at 765 nm after incubation at room temperature for 1 h. Gallic acid was used as a reference standard, and the results were expressed in milligrams of gallic acid equivalents (GAE)/liter of milk.

The analysis of the phenolic profile was carried out in agreement with [34,35,36]. In brief, milk samples (1 g each) were extracted in a solution of 15 mL of 1% formic acid in 80:20 methanol/water solution (LCMS grade, VWR, Milan, Italy) at room temperature using an Ultra-Turrax apparatus. The phenolic compounds were then screened in the extracts using UHPLC liquid chromatography (Agilent 1290 series) coupled to a quadrupole-time-of-flight mass spectrometer (G6550 iFunnel), [35]. Deconvolution and compounds annotation were carried out using the software Profinder B.06 (from Agilent Technologies), based on the database exported from Phenol-Explorer 3.6 [34]. Both monoisotopic accurate mass and isotopic profile (i.e., isotope spacing and ratio), together with 5 ppm tolerance for mass accuracy, were used for identification. The identification was carried out according to Level 2 (putatively annotated compounds), as set out by the COSMOS Metabolomics Standards Initiative (http://cosmos-fp7.eu/msi (accessed on 25 September 2020)). To gain semi-quantitative data, calibration curves were prepared from standard solutions of single pure phenolics (Extrasynthese, Lyon, France), as previously reported [36]. Ferulic acid (for hydroxycinnamic acids and other phenolic acids), matairesinol (for dibenzylbutyrolactone and dihydroxydibenzylbutane lignans), sesamin (furan and furofuran lignans), cyanidin (anthocyanins), catechin (flavanols), luteolin (flavones and other remaining flavonoids), resveratrol (stilbenes), 5-pentadecylresorcinol (alkylphenols) and tyrosol (tyrosols and other remaining phenolics) were used with this purpose. The classes of phenols were calculated as follows: flavonoids (flavonoids + luteolin), ferulate (ferulic acid + tyrosol), non-flavonoids (ferulate + sesamin).

At the same time as the milk sampling, randomized samples of the herbage from grazed pasture (samples of the pasture on offer) and supplied feedstuffs were sampled in each farm and stored at −20 °C until their chemical analysis. DM was determined by oven-drying at 105 °C to constant weight (AOAC, 2005; ref. 934.01), organic matter and total ash by muffle furnace (AOAC, 2005; ref. 942.05), crude protein (CP) by the Kjeldahl method (AOAC, 2005; ref. 976.05) and ether extract (EE) by Soxhlet analysis (AOAC, 2005; ref. 2003.05). In addition, the starch was measured with a polarimetric method according to the European Commission (1999). The NDF (neutral detergent fiber assayed with a heat stable amylase and expressed exclusive of residual ash), ADF (acid detergent fiber, expressed exclusive of residual ash) and ADL (acid detergent lignin) were determined [37] using an Ankom 220 Fiber Analyzer equipment (Ankom Technology, New York, NY, USA). The powdered material was used for the extract preparation of the total phenolics [38]. The total phenolics, were analyzed using Folin–Ciocalteau reagent and expressed as a tannic acid equivalent (mg TAE g^−1^ DM) according to procedures previously described [38]. Briefly, the samples (0.5 g each) were incubated in a solution of acidified methanol and then filtered. The filtrate was added to a solution including Folin-Ciocalteau reagent and sodium carbonate. After two hours, the samples were assayed for total polyphenols by a spectrophotometer (765 nm) using tannic acid as a standard. To measure non-tannic polyphenols the same procedure was used but the filtrate was pretreated with methyl-cellulose and ammonium sulphate before adding the Folin-Ciocalteau reagent. Tannic phenolics were measured as the difference between total and non-tannic polyphenols.

### 2.3. Statistical Analysis

Prior to statistical analysis, the data on FA composition were processed to calculate the following FA classes: short chain fatty acids (SCFA C4:–C11:0); medium chain fatty acids (MCFA C12:–C17:1); LCFA (C18:0–C26:0); odd and branched chain FA (OBCFA = isoC13:0 + anteisoC13:0 + isoC14:0 + isoC15:0 + anteisoC15:0 + C15:0 + isoC16:0 + isoC17:0 + anteisoC17:0 + C17:0 [39]; monounsaturated fatty acids (MUFA all FA with single double bond); PUFA (all FA with more than one double bond); saturated fatty acids (SFA all FA without double bonds); unsaturated fatty acids (UFA all FA with one or more double bonds). Several classes of phenols were also calculated as flavonoids (cyanidin + luteolin), ferulate (ferulic acid + tyrosol) and non-flavonoids (ferulic acid + tyrosol + sesamin).

Data for macro-composition (Fat, protein, casein, Lactose and SCC), 68 FA (expressed as % total FAME), 7 phenols, 1 total gallocatechin equivalent, FRAP Vitamin A and E, Cholesterol, DAP and colour (L *, a * and b *) were analyzed with PCA SAS PRINCOMP procedure (SAS Institute Inc., Cary, NC, USA).

A principal component analysis (proc princomp performed by SAS) was used to reduce the original number of variables to a new set of fewer variables (principal components, PC). The number of PCs retained was defined according to the amount of explained variance and by adopting Kaiser’s Criterion (eigenvalue > than 1). The PCs loadings were interpreted as the coefficients of the linear combination with the original variables. Principal component scores (*n* = 42) were analyzed using the proc mixed model for repeated measurement (SAS) to investigate the structural and managerial studied factors affecting significant macro and micro milk composition. The means and significance of the factors were detected at *p* < 0.05 using Tukey test. In order to stress the effects of the different phenological stages, the data of GW and FW stages were considered in the same group as the “vegetative phase” and compared to the MS stage. In the same way, in the pasture botanical composition (BC), the occurrence of forbs and legumes species, less present in pasture, were considered in the same group (forbs) to compare with the grasses species group. The model adopted for repeated measurement analysis was:Yijhtmo = µ + HeI + PPSj + BCh + SDt + Fm + Ɛijhtmo(1)
where:

Yijhtm = is the studied variable; 

µ = is the mean;

HeI = is the fixed effect of the herbage intake (HeI, i = high, medium and zero);

PPSj = is the fixed effect of the phenological stage (PPS, j = GW/FW or MS stage);

BCh = is the fixed effect of plant botanical composition (BC = H = grasses or forbs);

SDt = is the repeated fixed effect of the sampling date (SD, t = D1, D2, D3 and D4);

Fm = is the random effect of farm (F, m = 1 to 11)

Ɛijhtmo = is the error term.

Total supplementation and stocking rates were not considered in the model since any significance with the PC scores was detected.

Finally, Person’s index correlation (r) was studied between the significant factors pointed out by repeated measurement analysis and the original milk composition variables with the higher loadings. Person’s index correlation (r) between the milk’s macro and micro chemical components that showed the highest loadings were also investigated.

## 3. Results and Discussion

### 3.1. Dairy Farming System and Feeding Managements

On average, the UAA was around 68 ha, and the flock size was 204 head of Sarda sheep (Table 1). The total supplementation amount (hay and concentrate) supplied by the sheepherders is a complement of pasture herbage diet contribution and it depends on pasture herbage availability, milk yield, and the animals’ physiological stage in terms of DIM as also reported by [8]. The MS plant stage led to a decrement of pasture palatability and nutritive value of the total daily diet intake as a consequence of the reduction in the leaves to steam ratio that led to an increase of the ADF as reported by [14]. The forage (herbage + hay)/concentrate ratio in the animal diet changed throughout the whole survey period, with the lowest values in January (70% of forage in the total diet) and the highest in May (more than 80% of forage in the total diet) when the herbage availability and sheep grazing activity were maximum. To outline, all of the sheepherders followed the same feeding strategy, using a higher quantity of hay and concentrate during the winter than in spring in response to the lack of herbage in the permanent pasture, and the higher amount of available pasture herbage or standing hay in spring and summer, respectively. This was also reported by [40] in similar extensive dairy sheep farms located in the Basque Country. The estimated pasture herbage intake and total supplementation ranged between 0 and 1777, and 100 and 1850 g DM/day/head, respectively, as reported in Table 1.

### 3.2. Grassland Botanical and Chemical Composition

The main representative forage species in pasture were grasses composing, on average, 65% pasture DM based, followed by forbs (15% DM) and legumes (10% DM), with different patterns of evolution from January to May. The frequency of the botanical families was similar to the results found by [7] in the same study area. The proportion of legumes and forbs increased, whereas grass species decreased in the pasture from January to May. During January and March, the plants were characterized by GW stages in agreement with their growing degree days linked to environmental drivers, such as temperature and rainfall [41]. In May, most plants were in the FW stage, whereas in July, most plants were in the MS stage without vegetative activity. The ether extract decreased from January to July as has also been reported previously [14]. The total phenols increased from January to May in parallel with the legumes and forbs pattern in the grassland, in agreement with [42]; a drop (30%) was observed in July compared to March (MS and GW, respectively) according to [38]. The pasture CP content significantly (*p* < 0.05) decreased from March (GW) to May (FW), passing from 16.69 to 13.16 CP% of DM mainly due to the natural drop of CP content in the grasses component; in addition, the ADF content increased mainly from March (GW) to May (FW) by 23% (*p* < 0.05). Overall, the observed reduction in terms of EE, CP and quality of fiber from vegetative to reproductive stage changes is probably more linked to the high value of the leaves to stems ratio during GW than the FW or MS stage, as also reported by [14]. The evolution pattern of the grasslands’ botanical composition and their phenological stage affect the chemical composition of pasture with different magnitude. In fact, it is known that in the Mediterranean basin, when plants turn from the vegetative to the reproductive stage, the worse effects on CP drop and NDF increase is for grasses compared to legumes, as also reported by Cabiddu et al. [10].

### 3.3. Milk Chemical Composition and Relationship with Structural and Managerial Factors

In previous years, different attempts were carried out to explore the datasets from experimental studies using the multivariate approach, to evaluate the association between feeding management and milk fatty acid profiles [19,20,21]. In this study, we consider the “on farm” level additional factors that are not common in the bibliography, such as the natural pasture’s botanical composition and the estimated animal intake of herbage, hay and concentrate. However, the total number of observations is not very large (*n* = 42) due to the difficulty in collecting the data for such a wide number of variables.

Results from PCA showed that 16 out of 101 PC were found to be able to explain about 92% of the total variance of the system. The variance explained ranged from 37.00% for PC1 to about 1.00% for PC16. Only the first five components were detailed in this paper that explained around 70% of the total variance (Table 2). The five components (PCs) were chosen considering a threshold of 0.13 (absolute value) of eigenvalues.

About 90% of the milk FAs loaded on PC1–PC5. The overall pattern of milk FA composition includes 26 SFA and 42 UFA (21 MUFA and 21 PUFA) as reported in Table 1. The average values of the milk FA content confirmed, partially, the previous reports on extensive dairy sheep farm systems located in a few areas of Sardinia with different forage crops and a permanent pasture-based diet [11,43], whereas the milk total n-3 FAs (Table 1) appeared lower when compared to the milk of sheep grazing on crops [44].

SFA, MUFA and PUFA accounted for almost 68%, 25% and 7% of the total FAs, respectively. Palmitic, oleic, and linoleic acid are the main SFA, MUFA, and PUFA, respectively, in agreement with results from [43] in sheep, and [19] in cow’s milk.

The PC1, (variance explained 37%) presented the highest positive loadings for lactose, de novo FA synthesis (SCFA) and biohydrogenated FA intermediates, whereas OBCFA had negative loadings values (Table 2). Figure 1 represents the plot of loadings between PC1 and PC2 and shows a high correlation between the milk yield (MY) and lactose (loadings were 0.118 and 0.136, respectively), in agreement with [45], which underlined the role of milk lactose as an indicator of energy balance intake in line with the distribution of oleic acid (C18:1 c9) located on the opposite side of lactose and MY. Therefore, as expected, animals in a positive energy balance could increase MCFA and SCFA in milk as also found by [46,47], according with the high availability of acetate and 3-hydroxybutyrate, which are the precursors of lipid de novo synthesis at the mammary level.

These results partially agree with [48,49], who found a strong link between herbage intake and cow’s milk and lamb’s meat fatty acid composition, respectively. Loadings detected for SCC between the PC1–PC2 showed low values, and this agrees partially with [50], who observed that a pasture-based diet did not affect milk SCC. The high correlation between SCFA and lactose (0.78; *p* < 0.01), SCFA and MY (r = 0.68; *p* < 0.01), and lactose and MY (r = 0.76; *p* < 0.01) pointed out the effect of a high energy intake, associated with the high availability of acetate and 3-hydroxybutyrate, both contribute to increase milk SCFA by de novo synthesis [51], in agreement with [19,21,45], and in line with the previous studies in cows [19], buffalo [52], and in sheep [21]. The significant effect (*p* < 0.02) of herbage intake (HeI) on the PC1 scores (Table 3) probably partially explains the above results, and the position in Figure 1 of lactose, MY and de novo FA change, along the PC1 axis; in particular, milk total SCFAs are positively correlated to fresh herbage intake levels (r = 0.73; *p* < 0.01). A significant effect of plant PS on PC1 scores was detected too (Table 3), that could be due to an increase of the herbage ADF/NDF ratio (from vegetative to reproductive plant stage) with an implication for rumen microbial activity and, consequently, on milk OBCFA, as also reported by [39] in agreement with the positive correlation (r = 0.86; *p* < 0.01) between OBCFA milk content and plant phenological stage.

Plant PS is a key driver of animal metabolism, effecting changes on ruminal and milk components, in particular for FA, as also reported by [53]. A negative correlation was found between milk SCFA content and PS (r = −0.82; *p* < 0.01), with a higher level seen when sheep were grazing plants at the GW stage, compared to the FW stage, when fiber digestibility decreased [54], and with a detrimental effect on FA precursor availability for de novo synthesis at the mammary level. The change in the diet’s botanical composition (from grass to legumes and forbs) also affected the PC1 score (*p* < 0.01), probably as a consequence of plant secondary metabolites (PSM), which occur in legumes and forbs plants species more than in grass species [55]. As is well known, PSM in the diet could affect ruminal microbial activity with several effects on the milk’s FA profile and other micro-components, likewise phenols. In this study, we found a low negative value relationship between milk LCFA and the occurrence of grass in pasture (r = −0.33; *p* < 0.03) with no clear explanation, probably due to their low loading value for PC1. In addition, no effect on milk phenol profiles were detected when fresh pasture contribution in the diet increased. Increased legumes and forbs are also well represented in Figure 1, where phenols are mainly located near the origin of the axis.

The PC2 (variance explained 16.1%) had a negative loadings values for SFA and MCFA, whereas positive loadings values were detected for DAP, LCFA and UFA, and MUFA. No significant effect was detected between the PC2 scores and herbage intake, plant phenological stage or plant botanical composition (Table 4).

However, the positive correlation found between fresh herbage intake and milk linolenic acid content (r = 0.42; *p* < 0.01) partially explains the high and opposite loadings values of UFA and SFA, thanks to the plant’s PS (GW and FW) with a pivotal effect on the fatty acid precursor content in the herbage [14,53]. In addition, a good correlation was found between herbage intake and milk C18:2 t11c15 content (r = 0.60; *p* < 0.01) and between linolenic acid and C18:2 t11c15 milk content (r = 0.72; *p* < 0.01); it is well known that an increase in herbage intake leads to an increase in linolenic acid (ALA) and C18:2t11c15 in milk, as also reported by [56]. The BH intermediate (in particular, trans C18:1 isomers) with positive loadings were all related to the availability in the diet of linolenic acid (ALA), in agreement with [57], even if linoleic acid (LNA) and ALA showed low loadings values for PC2. A positive correlation between total trans C18:1 trans isomers (r = 0.70; *p* < 0.01) and milk ALA content was detected due to a higher intake of daily fresh pasture herbage, which increases ALA intake, as also reported by [20, 57] and is associated with an increase of total C18:1 trans [56]. In spite of any significant effect of hay or concentrate supplementation on PC scores, we found a positive correlation between hay intake and C15:0i, (r = 0.52; *p* < 0.01) as a probable consequence of the association effect of the concentrate. A high level of diet concentrate could lead to a detrimental effect on ruminal activity and OBCFA milk content, as also reported by [39]. In this case, it is very difficult to explain how the concentrate influences the milk component since no relationship was found between the forage to concentrate ratio and the OBCFA milk content. For instance, the C18:2 t9c12 appears positively correlated with the concentrate intake (r = 0.61; *p* < 0.01), and with the forage to concentrate ratio (r = 0. 32; *p* < 0.05) despite its low loadings values in both PC1 and PC2 (0.07 and 0.09, respectively).

The PC3 (variance explained 7.3%, Table 2) was characterized by high positive loadings values for ferulates non-flavonoids, total phenols, and CLA c9t11. A positive significant correlation was found between herbage intake and ferulates (r = 0.37; *p* < 0.05) and the CLA c9 t11 (r = 0.55; *p* < 0.01) in milk, respectively. This agrees with the positive correlation found between milk non-flavonoids, and ferulate with a n-3/n-6 ratio (r = +0.51 and 0.39; *p* < 0.05, respectively). Ferulates are the phenylpropanoid compounds that occur, in particular, on grass cell walls and in small amounts in legume cell walls as a main component of lignin [58]. These components are mainly responsible for ruminal fiber degradation, even if during the vegetative stage they are more degraded than in the plant at the reproductive stage. Between 10–50% of ingested lignin, following ruminal digestion, could be absorbed, as reported by [59]. These phenolic compounds, released at the ruminal level [60], showed a toxic microbial effect that could be the reason for the protective effect against the ruminal biohydrogenation of PUFA, which led to an increase in the n-3/n-6 ratio in the milk. This is also in line with the relationship found between ferulic acid and ALA (r = 0.41; *p* < 0.01) since linolenic acid represents the main fatty acid in fresh herbage. A negative correlation was found between CLA c9t11 and the plant’s phenological stage, when passing from the growing and flowering to the maturity stage, as it probably resulted in a decrease in the precursor [56].

The PC4 (variance explained 4.9%) is characterized by a positive loading for milk macro-composition (fat protein and casein) and C18:1 t10. Negative loadings were detected for ALA, EPA and DHA (see Table 2). A ruminants concentrate-based diet, with a low level of fiber, mostly exposes them to increases in trans 18:1 isomers and, in particular, C18:1 t10 [61]. In our study, no relationship was found between concentrate intake and total trans C18:1 fatty acid contents in milk, probably because of the high F/C ratio detected through the whole lactation period. Moreover, in our case, the level of rapidly fermentable carbohydrates (RFCHO) (starch + sugar) are probably low through the whole period (below 40% DM intake), seriously affecting the level of biohydrogenated intermediates, in agreement with [62].

The PC5 (variance explained 4.7%) is characterized by high positive loadings for the a * and L * colours parameters, while the total flavonoids (sum of cyanidin and luteolin) showed negative loadings. As expected, the L *, a * and b * parameters (Table 1) are lower (in particular, the b * values) than the values found in grazing cows [13], in line with [63], probably because, in sheep, there is a higher conversion rate of β-carotene to Vit. A than in cows [63]. Considering previously results found in grazing (4 h/day) sheep on natural pasture [64], we found high values of b * (+21%), whereas the L* and a* values were lower (−8% and −65%, respectively). The occurrence in the permanent pasture of plant secondary metabolites as phenols could also be responsible for the low value of a * and the decrease in brightness (low value of L *). A slight positive correlation was found between milk fat and the colour parameters (r = 0.57, 0.54 and 0.46 for b *, a * and L *, respectively, *p* < 0.01), since milk carotenoids (lutein) are mainly located in the fat, as also reported by [65]. The permanent pasture-based diet could affect the intake of plant secondary metabolites [63] with subsequent interference upon ruminal metabolism. Color parameters are influenced by several molecules, such as carotenoids and phenols [65]. As expected, a little negative correlation was found between Vit. A and the b * parameter (r = −0.35 *p* < 0.05). Considering the phenols profile, only sesamin appeared to be positively correlated with b* (r = +0.34; *p* < 0.02); sesamin belongs to the lignans component and is characterized by being able to be absorbed at 550 nm near the yellow region, as reported by [66]. As such, it is probable that it is not only carotenoids that could affect the b * parameter in sheep milk. Even if difficult, considering the results from the recent study, we assume that our results partially agree with [67], who found effects on the cheese colour parameter when the authors compared pasture vs. indoor feeding, although the % of legumes and forbs detected in the pasture was high. In addition, more herbage in the diet is positively correlated with n-3 FAs in the milk’s content. Moreover, it appears that flavonoids correlate negatively with total n-3 FAs (r = 0.30; *p* < 0.05) and positively with total n−6 FAs (r = 0.29; *p* < 0.05).

## 4. Conclusions

This study focuses at the farm level in an attempt to better evaluate the relationship between a few farm management choices and milk quality in grazing dairy sheep reared on permanent grassland. Moreover, it provides a tool by which to transfer research knowledge to farmers (even if the numbers of observations were restricted compared to the total variables considered) and provides new suggestions for research models. The fresh herbage intake and plant phenological stage appear to be the main drivers for OBCFA and SCFA and lactose content. Considering the other micro components in milk, this study pointed out the negative relationship between total supplementation and milk phenols content, in particular ferulates and non-flavonoids. The relationship between milk colour parameter (b *) and sesamin, and between the milk phenols and colour parameters could be considered in an authentication procedure to link animal products with the territorial brand, although this needs to be confirmed in experimental conditions. Our basal hypotheses are only partially explained by these results. Regarding the first hypothesis, the dataset exploration needs to be increased, taking into account the effect of years and to validate the results. For the second hypothesis, the methodological approach evaluated is a promising challenge to study the effect of farming practices on several traits of milk sheep reared on permanent grassland. In conclusion, even if these preliminary results are derived from commercial farms, they could help to establish future threshold values of biomarkers in milk sourced from grazing dairy sheep with natural, permanent pasture-based diets. These results need to be confirmed at the farm level using a larger database.

## Figures and Tables

**Figure 1 animals-12-02675-f001:**
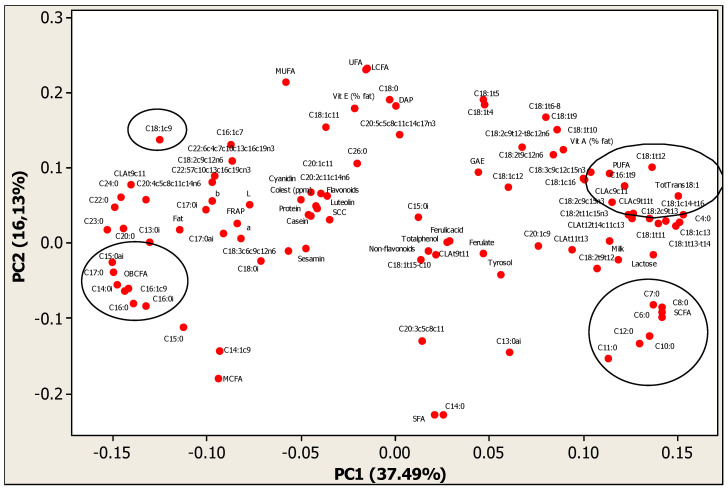
Plot of the loadings of the first two principal components (PC1 and PC2; variance explained 37.49 and 16.13, respectively).

**Table 1 animals-12-02675-t001:** Descriptive statistics for farm structure, husbandry, managerial factors, and milk traits.

	*n*	Average	SEM	Min	Max
Flock (n. sheep)	42	204.143	26.388	20	700
UAA (Ha)	42	67.714	6.376	14	132
Sampling date (SD)	42	2.548	0.174	1	4
Plant phenological stage	42	1.262	0.069	1	2
Herbage intake (g DM head^−1^ day^−1^)	42	883.622	106.577	0	1777
Total supplementation (g DM head^−1^ day^−1^)	42	566.500	73.723	100	1850
Concentrate intake (g DM head^−1^ day^−1^)	42	381.690	33.376	100	1300
Hay intake (g DM head^−1^ day^−1^)	42	184.810	53.059	0	1350
Milk yield (l head^−1^ day^−1^)	42	0.879	0.058	0.32	1.8
Milk fat (%)	42	6.184	0.134	3.73	7.88
Milk protein (%)	42	5.409	0.051	4.66	6.07
Milk lactose (%)	42	4.604	0.040	3.92	4.95
SCC	42	1530.905	143.389	236	3959
Casein (%)	42	4.131	0.045	3.47	4.72
Milk vit A (% fat)	42	1.337	0.040	0.904	1.878
Milk vit E (% fat)	42	3.983	0.278	1.569	11.450
Cholesterol (ppm)	42	211.020	8.160	121.070	453.810
DAP	42	10.281	0.568	4.249	25.441
FRAP (µmol L^−1^ FeSO_4_*7H_2_O)	42	191.901	18.659	12.3	498
b*	42	6.20	0.16	4.60	8.81
a*	42	−3.13	0.07	−3.88	−0.92
L*	42	70.39	0.52	62.38	76.68
Cyanidin (mg/L)	42	0.372	0.017	0.116	0.641
Luteolin (mg/L)	42	1.708	0.100	0.383	3.074
**Flavonoids** (mg/L)	42	2.080	0.117	0.499	3.716
Ferulic acid (mg/L)	42	14.636	1.016	5.236	30.322
Tyrosol (mg/L)	42	15.468	0.439	9.665	21.448
**Ferulate** (mg/L)	42	30.104	1.123	20.127	50.301
Sesamin (mg/L)	42	29.641	0.574	22.247	39.545
**Non-flavonoids** (mg/L)	42	59.932	1.351	43.554	81.090
TP (mg/L)	42	62.012	1.342	45.746	83.801
GAE (mg/L)	42	329.603	21.426	70	879
SCFA					
C4:0 (% FAME)	42	4.056	0.059	3.373	4.731
C6:0	42	2.709	0.095	1.599	3.576
C7:0	42	0.025	0.002	0.006	0.053
C8:0	42	2.174	0.102	1.061	3.221
C10:0	42	6.335	0.310	2.669	9.883
C11:0	42	0.303	0.012	0.132	0.454
MCFA					
C12:0 (% FAME)	42	3.388	0.137	1.808	4.977
C13:0i	42	0.031	0.002	0.015	0.061
C13:0ai	42	0.038	0.001	0.028	0.052
C14:0i	42	0.157	0.009	0.048	0.297
C14:0	42	9.956	0.167	6.550	11.851
C14:1 c9	42	0.180	0.007	0.110	0.294
C15:0i	42	0.075	0.002	0.059	0.105
C15:0ai	42	0.318	0.014	0.181	0.536
C15:0	42	1.185	0.032	0.634	1.675
C16:0i	42	0.382	0.014	0.202	0.601
C16:0	42	24.963	0.487	21.198	33.062
C16:1 t9	42	0.200	0.013	0.062	0.390
C16:1 c9	42	0.829	0.031	0.574	1.322
C16:1 c7	42	0.289	0.012	0.075	0.445
C17:0	42	0.783	0.027	0.557	1.255
C17:0i	42	0.505	0.010	0.380	0.618
C17:0ai	42	0.562	0.016	0.298	0.764
LCFA					
C18:0 (% FAME)	42	9.268	0.246	6.481	13.939
C18:0i	42	0.074	0.003	0.041	0.118
C18:1 t4	42	0.013	0.002	0.004	0.071
C18: t5	42	0.015	0.002	0.004	0.069
C18:1 t6 + t8	42	0.251	0.017	0.116	0.793
C18:1 t9	42	0.282	0.011	0.191	0.611
C18:1 t10	42	0.416	0.030	0.174	0.952
C18:1 t11	42	2.378	0.163	0.617	4.736
C18:1 t12	42	0.446	0.021	0.204	0.711
C18:1 t13 + t14	42	1.150	0.071	0.364	2.128
C18:1 c9	42	17.015	0.544	10.586	25.075
C18:1 t15 + c10c	42	0.364	0.035	0.075	1.216
C18:1 c11c	42	0.387	0.009	0.300	0.556
C18:1 c12	42	0.199	0.010	0.118	0.402
C18:1 c13	42	0.096	0.004	0.048	0.143
C18:1 c14 + t16	42	0.552	0.025	0.224	0.763
C18:2 t9t12	42	0.042	0.006	0.004	0.155
C18:2 c9t13	42	0.492	0.024	0.201	0.804
C18:2 c9t12t8c12n6	42	0.174	0.009	0.066	0.300
C18:1 c16	42	0.130	0.005	0.079	0.230
C18:2 t9c12n6	42	0.027	0.003	0.006	0.078
C18:2 t11c15n3	42	0.427	0.037	0.053	1.175
C18:2 c9c12n6	42	2.035	0.057	1.285	2.957
C18:2 c9c15n3	42	0.020	0.002	0.003	0.055
CLA c9t11	42	1.498	0.079	0.619	2.677
CLA t9c11	42	0.110	0.005	0.064	0.170
CLA c9c11	42	0.057	0.005	0.011	0.142
CLA t12t14c11c13	42	0.027	0.003	0.000	0.131
CLA t11t13	42	0.044	0.006	0.000	0.276
CLA t9t11	42	0.029	0.001	0.020	0.057
C18:3 c6c9c12n6	42	0.035	0.001	0.011	0.050
C18:3 c9c12c15n3	42	1.029	0.040	0.339	1.698
C20:0	42	0.342	0.025	0.178	0.820
C20:1 c9	42	0.009	0.001	0.000	0.018
C20:1 c11	42	0.037	0.002	0.015	0.057
C20:2 c11c14n6	42	0.018	0.001	0.008	0.030
C20:3 c5c8c11	42	0.063	0.012	0.000	0.283
C20:4 c5c8c11c14n6	42	0.143	0.006	0.096	0.259
C20:5 c5c8c11c14c17n3	42	0.074	0.002	0.050	0.127
C22:0	42	0.167	0.008	0.090	0.293
C22:5 c7c10c13c16c19n3	42	0.170	0.005	0.116	0.275
C22:6 c4c7c10c13c16c19n3	42	0.064	0.004	0.032	0.133
C23:0	42	0.071	0.004	0.037	0.142
C24:0	42	0.077	0.004	0.037	0.143
C26:0	42	0.044	0.002	0.015	0.091

*n* = number of observation; FAME = fatty acid methyl ester; Flavonoids = (Cyanidin + luteolin); Ferulate = (ferulic acid + tyrosol); Non-flavonoids = (ferulic acid + tyrosol + sesamin); TP = total phenols; GAE = gallic acid equivalent; DAP = degree of antioxidant protection; FRAP = ferric reducing antioxidant power; SCFA = short chain fatty acids; MCFA = medium chain fatty acids; LCFA = long chain fatty acids.

**Table 2 animals-12-02675-t002:** **Eingenvectors,** eigenvalues, and percentage of variance explained of the first five principal components (PC) extracted from the correlation matrix.

Var	PC1	PC2	PC3	PC4	PC5
*Milk yield and macro-composition*					
Milk yield	0.118	−0.021	−0.127	−0.089	0.040
Milk fat	−0.115	0.018	0.114	0.174	0.027
Milk protein	−0.045	0.037	0.200	0.263	0.119
Milk lactose	0.136	−0.015	−0.065	0.026	0.014
SCC	−0.035	0.033	−0.065	−0.145	−0.037
Casein	−0.046	0.039	0.183	0.274	0.126
Milk vit A	0.083	0.119	−0.090	−0.085	0.138
Milk vit E	−0.022	0.180	0.173	−0.007	0.031
Cholesterol	−0.050	0.058	0.240	0.041	0.135
DAP	0.000	0.183	0.065	0.024	−0.052
FRAP	−0.084	0.028	0.018	−0.039	0.004
*Colours parameters*					
b *	−0.097	0.057	0.128	0.025	−0.090
a *	−0.082	0.007	0.086	0.083	0.162
L *	−0.077	0.051	0.067	0.084	0.196
*Phenols content*					
Cyanidin	−0.045	0.068	−0.123	0.052	−0.260
Luteolin	−0.042	0.047	−0.150	0.073	−0.241
**Flavonoids**	−0.042	0.050	−0.146	0.070	−0.245
Ferulic acid	0.027	0.003	0.307	−0.018	−0.049
Tyrosol	0.056	−0.041	0.065	−0.093	−0.117
**Ferulate**	0.046	−0.013	0.303	−0.052	−0.090
Sesamin	−0.048	−0.006	0.117	0.122	−0.114
**Non-flavonoids**	0.021	−0.015	0.303	0.006	−0.120
TP	0.018	−0.010	0.292	0.012	−0.143
GAE	0.044	0.095	0.115	0.019	0.063
*SCFA*					
C4:0	0.123	0.038	−0.098	−0.093	0.070
C6:0	0.141	−0.084	−0.040	−0.048	0.095
C7:0	0.136	−0.081	−0.015	0.123	0.070
C8:0	0.141	−0.091	−0.024	−0.027	0.085
C10:0	0.134	−0.123	0.002	0.005	0.068
C11:0	0.113	−0.153	0.019	0.035	0.054
*MCFA*					
C12:0	0.130	−0.132	0.019	0.033	0.052
C13:0i	−0.131	0.002	0.110	−0.094	0.017
C13:0ai	0.060	−0.145	0.064	0.099	0.066
C14:0	0.025	−0.228	0.054	0.024	0.008
C14:0i	−0.144	−0.063	0.040	−0.052	0.015
C14:1 c9	−0.093	−0.143	0.082	0.116	−0.033
C15:0	−0.112	−0.111	0.056	0.043	−0.022
C15:0i	0.012	0.035	−0.130	0.035	0.197
C15:0ai	−0.150	−0.038	0.034	−0.030	0.024
C16:0	−0.139	−0.080	0.017	0.023	−0.014
C16:0i	−0.132	−0.082	0.003	0.003	0.051
C16:1 c7	−0.088	0.132	−0.028	0.078	0.220
C16:1 c9	−0.142	−0.060	0.024	0.096	−0.005
C16:1 t9	0.121	0.076	0.136	−0.113	0.003
C17:0	−0.151	−0.025	0.021	0.038	0.001
C17:0i	−0.101	0.046	−0.012	0.003	0.131
C17:0ai	−0.092	0.013	−0.044	0.102	0.161
*LCFA*					
C18:0	−0.003	0.191	−0.074	−0.057	−0.032
C18:0i	−0.072	−0.022	0.085	−0.053	−0.101
C18:1 t4	0.047	0.184	−0.006	0.126	−0.089
C18: t5	0.046	0.192	0.017	0.134	−0.053
C18:1 t6 + t8	0.080	0.168	0.016	0.155	−0.079
C18:1 t9	0.086	0.152	0.027	0.158	−0.119
C18:1 t10	0.089	0.125	−0.059	0.197	0.054
C18:1 t11	0.139	0.027	0.104	−0.103	−0.050
C18:1 t12	0.136	0.101	−0.018	0.077	−0.029
C18:1 t13 + t14	0.151	0.028	−0.001	0.103	0.016
C18:1 c9	−0.125	0.138	−0.050	0.006	−0.056
C18:1 t15 + c10	0.014	−0.021	0.069	0.040	−0.118
C18:1 c11	−0.037	0.155	−0.028	0.077	0.115
C18:1 c12	0.060	0.074	−0.104	0.190	0.112
C18:1 c13	0.148	0.024	0.036	0.073	−0.034
C18:1 c14 + t16	0.153	0.039	−0.020	0.004	−0.007
C18:1 c16	0.099	0.087	−0.049	−0.043	−0.068
C18:2 t9t12	0.107	−0.034	0.002	0.003	−0.097
C18:2 c9t13	0.143	0.030	0.022	0.096	0.003
C18:2 c9t12t8c12n6	0.067	0.128	−0.039	−0.032	0.088
C18:2 t9c12n6	0.103	0.095	0.022	−0.106	0.152
C18:2 t11c15n3	0.135	0.033	0.096	0.021	0.060
C18:2 c9c12n6	−0.096	0.089	−0.125	−0.029	0.034
C18:2 c9c15n3	0.126	0.041	0.014	0.162	0.099
CLA c9t11	0.126	0.033	0.141	−0.116	−0.065
CLA t9c11	−0.141	0.078	0.021	0.050	0.051
CLA c9c11	0.115	0.055	0.162	−0.089	0.086
CLA t12t14c11c13	0.113	0.004	−0.002	0.108	0.095
CLA t11t13	0.094	−0.009	0.021	0.149	0.091
CLA t9t11	0.028	0.004	0.010	0.225	−0.149
C18:3 c6c9c12n6	−0.057	−0.009	−0.049	0.104	0.236
C18:3 c9c12c15n3	0.100	0.085	0.112	−0.156	0.037
C20:0	−0.144	0.021	−0.024	0.026	−0.092
C20:1 c9	0.076	−0.003	−0.029	0.174	−0.009
C20:1 c11	−0.039	0.067	−0.080	0.213	0.077
C20:2 c11c14n6	−0.037	0.063	−0.076	−0.077	0.105
C20:3 c5c8c11	0.014	−0.129	0.080	0.142	−0.085
C20:4 c5c8c11c14n6	−0.133	0.058	−0.034	−0.013	0.135
C20:5 c5c8c11c14c17n3	0.002	0.145	0.130	−0.185	0.169
C22:5 c7c10c13c16c19n3	−0.149	0.049	−0.008	−0.017	−0.043
C22:6 c4c7c10c13c16c19n3	−0.097	0.082	0.130	−0.133	0.129
C23:0	−0.087	0.110	0.108	−0.159	0.120
C24:0	−0.153	0.019	−0.004	−0.006	0.003
C26:0	−0.020	0.107	−0.003	−0.172	0.053
SCFA	0.141	−0.098	−0.021	−0.020	0.079
MCFA	−0.094	−0.178	0.044	0.037	0.010
LCFA	−0.015	0.232	−0.022	−0.024	−0.070
SFA	0.021	−0.228	−0.015	−0.004	0.083
UFA	−0.016	0.231	0.013	0.003	−0.074
MUFA	−0.058	0.214	−0.020	0.032	−0.090
PUFA	0.114	0.094	0.093	−0.080	0.025
OBCFA	−0.148	−0.055	0.023	0.030	0.045
totaltrans18:1	0.150	0.063	0.055	0.011	−0.033
Eigenvalue	37.87	16.13	7.30	4.93	4.44
Total Variance explained (%)	37.49	53.62	60.92	65.85	70.28

*n* = number of observation; Flavonoids = (Cyanidin + luteolin); Ferulate = (ferulic acid + tyrosol); Non-flavonoids = (ferulic acid + tyrosol + sesamin); TP = total phenols; GAE = gallic acid equivalent; DAP = degree of antioxidant protection; FRAP = ferric reducing antioxidant power; SCFA = short chain fatty acids; MCFA = medium chain fatty acids; LCFA = long chain fatty acids.

**Table 3 animals-12-02675-t003:** *p* values, standard error (SE), degree freedom (DF), and *t* value of PC1 estimated score values, performed by a GLM repeated measures procedure for the factors of herbage intake (HeI), plant phenological stage (PPS), and pasture botanical composition (BC).

FACTORS		Score PC1	SE	DF	*t* Value	*p*
HeI	High	−2.8344	1.1007	14	−2.58	0.0220
	Medium	−5.9794	1.5394	14	−3.88	0.0017
	Zero	−2.6156	0.9691	14	−2.70	0.0173
PPS	GW/FW	2.7864	0.7330	10	3.80	0.0035
	MS	−10.4060	1.4720	10	−7.07	<0.0001
BC	Grass	−4.2524	0.7118	10	−5.97	0.0001
	(forbs + legumes)	−3.3672	0.8183	10	−4.11	0.0021

**Table 4 animals-12-02675-t004:** *p* values, standard error (SE) degree freedom (DF), and *t* value of PC2 estimated score values, performed by a GLM repeated measures procedure for the factors of herbage intake (Hel), plant phenological stage (PPS) and pasture botanical composition (BC).

FACTORS		Score PC2	SE	DF	*t* Value	*p*
HeI	High	−1.1492	1.6990	14	−0.68	0.5098
	Medium	0.5833	2.3762	14	0.25	0.8096
	Zero	0.4316	1.4959	14	0.29	0.7772
PPS	GW/FW	0.9079	1.1314	10	0.80	0.4409
	MS	−0.9974	2.2722	10	−0.44	0.6700
BC	Grass	1.2054	1.0988	10	1.10	0.2983
	(forbs + legumes)	−1.2949	1.2632	10	−1.03	0.3294

## Data Availability

The data presented in this study are available on request from the corresponding author.

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
