# Peer review of "Dairy Sheep Grazing Management and Pasture Botanical Composition Affect Milk Macro and Micro Components: A Methodological Approach to Assess the Main Managerial Factors at Farm Level"

_animals, 2022, doi:10.3390/ani12192675_

Round 1

Reviewer 1 Report

Manuscript number Animals-1931250

Title Dairy sheep grazing management and pasture botanical composition affect milk macro and micro components : a methodological approach to assess the main managerial factors at farm level

Overall comments,

Many researchers were interested in the assessment of a data methodological technique in a case study at the commercial farm level, to aggregate a high number of variables observed in milk sheep, into fewer latent structures. This work is intriguing, however, after close reading, several issues were resolved, necessitating revision.

Abstract

General

-

Introduction

General

-There is a lack of connection between paragraphs

-There is a lack of hypothesis

-There are many several previous studies about the assessment of a data methodological approach in a case study at the commercial farm level. Therefore, what is the research gap from this study obtained?

-What is the new knowledge from this study?

Specific

            -

Materials and method

General

-There is a lack of experimental design.

- Please explain the statistical differences for the p-value.

Specific

-L177-180: Please provide a more detailed analysis of Vitamin A, E, and Cholesterol.

-L181-182: Please provide a more detailed analysis of FRAP and Gallic acid

Results and Discussion

General

            -

Specific

            -L308: (+23%; 307 P < 0.05).. >> (+23%; 307 P < 0.05).

            -L293-308: Please discuss more in this parameter.

Conclusion

General

·       You should recommendation for further research

Reference

·       Modify format according to Journal style

Author Response

Dear referee,

many thanks for all of your comments. We agree with all of your suggestion. Please see below in red our response and in the attached file the new version of paper with the changes based on your comments.

Overall comments,

Many researchers were interested in the assessment of a data methodological technique in a case study at the commercial farm level, to aggregate a high number of variables observed in milk sheep, into fewer latent structures. This work is intriguing, however, after close reading, several issues were resolved, necessitating revision.

AU: the authors thanks a lot the reviewer for  all the suggestion

Abstract

General

-

Introduction

General

-There is a lack of connection between paragraphs

AU: authors made a change along the introduction to better connect the paragraph

-There is a lack of hypothesis

AU: the hypothesis are added at the end of introduction

-There are many several previous studies about the assessment of a data methodological approach in a case study at the commercial farm level. Therefore, what is the research gap from this study obtained?

AU: as reported in the introduction in the bibliography there are rarely study on permanent grassland (more studies refer to forage grazed crops!) with a “few” number of farm where sometimes authors report botanical composition and plant phenological stage connected to milk composition. In addition to our knowledge no study occur in the bibliography with a large range of variables like our study  considering milk colours vitamins, macro composition, fatty acids profile FRAP, phenols, and DAP. In addition several case study report the data using MIRS/NIRS technology which for some FAs are difficult to compare with manual analysis. The attempt to connect feeding supplementation, botanical and plant phenological stage  of grassland in 11 farms and using manual labs analysis requires very expansive cost.

-What is the new knowledge from this study?

AU: Overall these results are the first attempt to transfer the knowledge from experimental to farm level. This study carried out at farm level the effect of fresh herbage intake on such milk biomarkers (considering plant botanical composition and phenological stage with feeding supplementation) summarizing them into a small number of variables. In addition is need to study the relation with such colour parameter because in the past we underestimated this parameters in milk sourced from pasture based diet. In the past we link the milk colour only to the carotenoids…probably others interferences affect colours and need to be explored better.

Specific

            -

Materials and method

General

-There is a lack of experimental design.

AU: DONE. In sub-paragraph 2.1 we explained the total number of observation considered. In the sub-paragraph 2.3 we added details about statistical analysis.

.

- Please explain the statistical differences for the p-value.

AU: DONE see 2.3 sub-paragraph

Specific

-L177-180: Please provide a more detailed analysis of Vitamin A, E, and Cholesterol.

AU: DONE 

-L181-182: Please provide a more detailed analysis of FRAP and Gallic acid

AU:  DONE

Results and Discussion

General

            -

Specific

            -L308: (+23%; 307 P < 0.05).. >> (+23%; 307 P < 0.05).

AU: DONE. we rephrase it

            -L293-308: Please discuss more in this parameter.

AU: DONE, we rephrase it

Conclusion

General

  • You should recommendation for further research

AU: DONE, authors add further raccomendation…

Reference

  • Modify format according to Journal style

AU: many thanks the reference format is now ok. DONE,

Reviewer 2 Report

Dear authors,

I reviewed the manuscript identified as animals-1931250. In my opinion, the conclusions of the conducted research are clear and result from the obtained research results. The material used for the research is sufficient, and the research methods have been selected appropriately. The arrangement of the figures and tables is appropriate and presents the obtained results very well. Discussing the results against the background of other authors is very detailed. The publications cited by the authors of the article are well selected. For the most part, the authors refer to the latest knowledge published in renowned scientific journals. I could not find any mistakes in the scientific aspect of the manuscript.

I congratulate the authors for their hard work and the nice presentation of their work.

However, the authors did not avoid a few mistakes, which I will list below:

- a few punctuation problems are present in the manuscript. I suggest the Authors double-check the text;

- not all acronyms are specified at the first mention (eg PCA on line 109).

In addition, I wanted to point out to the authors how the seasonal variability of feeding strategies highlighted by the authors is not common, as stated, only to extensive farms in the Basque country, but also to sheep flocks closely to the study area described in the manuscript. Therefore, I invite the authors to update this part of the discussion of the results. In this context, the following manuscript https://doi.org/10.3390/foods9081091, along with others, could be usefully included for this purpose.

Regards

Author Response

Dear referee,

many thanks for the suggestion to improve our manuscript.

See below our answer (in red!) and in the attached file the new vesrion of the paper ammended based on your suggestion. 

Dear authors,

I reviewed the manuscript identified as animals-1931250. In my opinion, the conclusions of the conducted research are clear and result from the obtained research results. The material used for the research is sufficient, and the research methods have been selected appropriately. The arrangement of the figures and tables is appropriate and presents the obtained results very well. Discussing the results against the background of other authors is very detailed. The publications cited by the authors of the article are well selected. For the most part, the authors refer to the latest knowledge published in renowned scientific journals. I could not find any mistakes in the scientific aspect of the manuscript.

AU: authors thanks the referee for hers/his comments.

I congratulate the authors for their hard work and the nice presentation of their work.

AU: authors thanks the referee

However, the authors did not avoid a few mistakes, which I will list below:

- a few punctuation problems are present in the manuscript. I suggest the Authors double-check the text;

AU: DONE

- not all acronyms are specified at the first mention (eg PCA on line 109).

AU: DONE , authors check the all document

In addition, I wanted to point out to the authors how the seasonal variability of feeding strategies highlighted by the authors is not common, as stated, only to extensive farms in the Basque country, but also to sheep flocks closely to the study area described in the manuscript. Therefore, I invite the authors to update this part of the discussion of the results. In this context, the following manuscript https://doi.org/10.3390/foods9081091, along with others, could be usefully included for this purpose.

AU: DONE, authors agree with the referee comment about the role of season (in particular plant phenological  but also botanical composition!!) on the milk and dairy products quality composition. We add the reference suggested by the referee even if the paper refer to cheese… but of course we appreciate a lot your suggestion  because all milk from sheep is processed into cheese.

Round 2

Reviewer 1 Report

The authors revised accordingly to my previous comments, however, there are minor concerned that were found and required the authors to consider. Please see in Manuscript file with comments.

Author Response

Dear referee

authors thanks you again for the suggestion to improve the paper.

We accept all the suggestion which you ask us. We hope that all fir your request

Best regards

Andrea
